# Solving complex nanostructures with ptychographic atomic electron tomography

Philipp M. Pelz [1,2,3] ✉, Sinéad M. Griffin [3,4], Scott Stonemeyer[4,5,6,7], Derek Popple [4,5,6,7], Hannah DeVyldere [2], Peter Ercius [3], Alex Zettl [2,4,5,7], Mary C. Scott [2,3,4] & Colin Ophus [3] ✉

Transmission electron microscopy (TEM) is essential for determining atomic scale structures in structural biology and materials science. In structural biology, three-dimensional structures of proteins are routinely determined from thousands of identical particles using phase-contrast TEM. In materials science, three-dimensional atomic structures of complex nanomaterials have been determined using atomic electron tomography (AET). However, neither of these methods can determine the three-dimensional atomic structure of heterogeneous nanomaterials containing light elements. Here, we perform ptychographic electron tomography from 34.5 million diffraction patterns to reconstruct an atomic resolution tilt series of a double wall-carbon nanotube (DW-CNT) encapsulating a complex ZrTe sandwich structure. Class averaging the resulting tilt series images and subpixel localization of the atomic peaks reveals a $Zr_{11}Te_{50}$ structure containing a previously unobserved $ZrTe_2$ phase in the core. The experimental realization of atomic resolution ptychographic electron tomography will allow for the structural determination of a wide range of beam-sensitive nanomaterials containing light elements.

Knowledge of the three-dimensional atomic structure of natural and manufactured materials allows us to calculate their physical properties and deduce their function from first principles. Because of this, methods for atomic structure determination have been a key area of research in the biological and physical sciences, including x-ray crystallography[1] and nuclear magnetic resonance spectroscopy[2]. For crystalline samples, micro-crystal electron diffraction provides data of similar quality to X-ray diffraction for solving structures[3]. More recently, cryo-electron microscopy has become the dominant method for atomic structure determination of molecules, either from ensembles fulfilling the single-particle assumption[4], or by cryo-electron tomography and subsequent subtomogram averaging[5]. However, these methods require averaging of thousands of near-identical structures. One method which is capable of solving structurally and chemically heterogeneous nanostructures is atomic electron tomography (AET) using scanning transmission electron microscopy[6–11]. However, the dark field imaging method used in these AET studies produces very little contrast for light elements, and requires too much electron dose to be used for beam-sensitive samples (ref. 12 and Supplementary Fig. 1). The predominant method to increase the contrast of weakly-scattering elements is phase-contrast microscopy. In plane-wave TEM, phase-contrast is typically introduced by recording images at varying defocus values of the objective lens and using iterative algorithms to reconstruct the phase-shift of the sample, or contrast transfer function correction of an image taken at a single defocus. Algorithms for three-dimensional phase-contrast

[1]Institute of Micro- and Nanostructure Research (IMN) & Center for Nanoanalysis and Electron Microscopy (CENEM), Friedrich Alexander-Universität Erlangen-Nürnberg, IZNF, 91058 Erlangen, Germany. [2]Department of Materials Science and Engineering, University of California Berkeley, Berkeley, CA 94720, USA. [3]The Molecular Foundry, Lawrence Berkeley National Laboratory, Berkeley, CA 94720, USA. [4]Materials Sciences Division, Lawrence Berkeley National Laboratory, Berkeley, CA 94720, USA. [5]Kavli Energy NanoSciences Institute at the University of California at Berkeley, Berkeley, CA 94720, USA. [6]Department of Chemistry, University of California at Berkeley, Berkeley, CA 94720, USA. [7]Department of Physics, University of California at Berkeley, Berkeley, CA 94720, USA. ✉e-mail: philipp.pelz@fau.de; cophus@gmail.com

tomography that simultaneously solve the multiple-scattering problem have been proposed[13,14], and demonstrated at sub-nm resolution[15].

In STEM, phase contrast can be achieved in multiple ways by recording momentum-resolved information of the scattered electrons. The annular bright-field mode employs an integrating annular detector and provides enhanced contrast for light elements, albeit with a strong thickness and defocus dependence[16]. Bright-field STEM is used to image whole vitrified cells[17,18]. Integrated differential phase-contrast, integrated center-of-mass imaging[19], and optimum bright-field imaging[20] are modern techniques that employ a segmented bright-field detector. Their simple processing provides real-time feedback to the user on segmented detectors as well as modern pixelated detectors[21]. Ptychography is a computational phase-contrast method that reconstructs a two-dimensional image from a series of full-field diffraction patterns recorded by scanning a confined beam over the sample. It can be implemented as a direct phase-retrieval method[22–24] or using iterative phase-retrieval algorithms[25,26] using many different experimental configurations[27]. Direct ptychography methods enjoy a low computational complexity and can provide real-time feedback when implemented on modern hardware[28], while iterative ptychography algorithms allow to jointly solve for the complex sample transmission function[29], a non-parametric probe wave function[30,31], sub-pixel scan positions[32,33], partial coherence effects in the experiment[33,34], fluctuating illumination during the scan[35] and can provide superior resolution and reconstruction quality under certain conditions[33,36]. Neural-network-based reconstruction methods improve the aforementioned direct methods while retaining real-time reconstruction capabilities[37,38]. The most popular implementation of ptychography in the TEM uses the STEM configuration in combination with a fast-framing direct electron detector positioned in the far field of the illuminated sample.

Electron ptychography is particularly attractive for its dose-efficiency and has been used to image beam-sensitive materials such as metal-organic frameworks[39], perovskites[40], and biological materials[41,42].

Multi-slice ptychography[43,44], a generalization of iterative ptychographic algorithms to reconstruct samples thicker than the depth-of-field, can be used to obtain three-dimensional information[45] from a single view and also increase the transverse spatial resolution[46]. Simulations show that point defects and interstitial atoms in crystalline materials should be resolvable in three dimensions from a single view with an axial resolution of a few nm with multi-slice electron ptychography[46]. To obtain higher axial resolution than possible from a single view, the sample has to be rotated around an axis non-parallel to the beam direction in a tomography experiment.

Ptychographic tomography originated in X-ray microscopy[47] and has developed into a standard method for three-dimensional nano- and microscale characterization at modern synchrotrons due to its high resolution and sensitivity. Ptychographic atomic electron tomography (PAET) has been proposed in simulations as a means to resolve light atoms in nanostructures at the atomic scale[12] and resolve the multiple scattering problem[44]. Low-dose pytchographic electron tomography was demonstrated at nanometer resolution in the TEM[48] for imaging of organic-inorganic hybrid nanostructures.

Low-dimensional van der Waals (vdW) materials such as the transition metal di- and tri-chalcogenide families or CNTs exhibit a range of desirable properties that often emerge as the material thickness reaches the one- or two-dimensional thickness limit[49–51]. In order to synthesize otherwise unstable vdW materials, encapsulation inside CNTs has been developed as a strategy to stabilize quasi one-dimensional structures[52–54] of transition metals. The encapsulation approach protects the interior structure from oxidation and can result in chiral structures[51,53] or fillings with high aspect ratio[55] that exhibit properties differing drastically from their bulk counterparts. While

electron microscopy has played an important role in the identification and characterization of the encapsulated phases, the detailed 3D structure of the interior nanowire is not always clear. For example, the encased material's structure often exhibits a dependence on the diameter of the nanotube[52], and can form complex three-dimensional structures inside the CNT, such as core-shell structures[56] and multi-layer moiré structures[57]. In the latter cases, the atomic structure cannot be uniquely determined from a single projection, and three-dimensional imaging is paramount for structure-function determination.

Here we experimentally demonstrate atomic structure determination of a complex double-wall CNT-encapsulated $Zr_{11}Te_{50}$ structure using atomic resolution PAET combined with unit cell averaging and DFT calculations.

## Results
### Experiment

The experimental setup for PAET in the STEM is shown in Fig. 1a. A converged electron probe is raster-scanned over the sample, with one diffraction pattern recorded at every probe position using a fast-framing direct electron detector operating at 87 kHz. The nanotube is tilted around its axis using a specialized dual-axis tomography holder, and at each tilt angle the scanning diffraction measurements are repeated. We reconstruct the partially coherent illumination wave function from each four-dimensional dataset, where the dominant mode of an example probe's complex wave function is shown in Fig. 1b. The probe wavefunctions and their positions are reconstructed jointly with the sample object wave. This feature of modern ptychographic reconstruction algorithms is especially beneficial to reduce the total dose in atomic resolution experiments because precise focusing before the acquisition is not necessary using PAET. Fig. 1c shows the phase of the object exit wave from three of the experimental projections. The additional crystalline contrast left of the filled DW-CNT stems from an adjacent unfilled triple-wall CNT, which we have masked from the reconstruction (See Supplementary Fig. 2 for the full field of view). The strong contrast of the carbon atoms in the 2D projections allows us to determine a zig-zag nanotube configuration and the dimensions of the semi-minor and semi-major axes of the elliptical DW-CNT (see Methods Section G), information unobtainable from the ADF signal at the same electron dose.

To increase the signal-to-noise ratio of the phase-contrast projections, and enable determination of the 3D structure of the center of the tube even with a limited experimental tilt range, we computed class averages along the nanotube, with three classes shown in Fig. 1d which correspond to the images in Fig. 1c. The size of these class images was chosen to include 11 repeats of the core structure and ~23 repeats of the DW-CNT structure. Figure 1e shows the average diffraction pattern of the 38 phase-contrast projections, which displays a high degree of periodicity along the tube. The $1/2.13$ Å$^{-1}$ reflection of the graphene lattice is clearly visible, as well as the 1st and 2nd diffraction order of the Zr-Zr and Te-Te lattice spacing of $1/3.95$ Å$^{-1}$ and $1/1.975$ Å$^{-1}$. The unit cell averaging procedure yielded a high-SNR image of the core structure, while the incommensurate 2.13 Å spacing of the graphene lattice in the carbon nanotube is highly suppressed.

### Reconstruction

We have reconstructed the 3D volume shown in Fig. 1f, from the 2D class images in shown in Fig. 1d, using the methods described in Methods Sec. E. From this reconstruction, we can clearly identify the outer two shells, which correspond to the DW-CNT, which have been colored in blue. We can also see the complex Zr-Te interior structure, colored in green and gold. This structure is highly periodic along the tube axis but does not possess a periodic crystal structure in the plane perpendicular to the nanotube axis. Instead, the Zr-Te structure possesses a distinct core and shell structure, with a high degree of

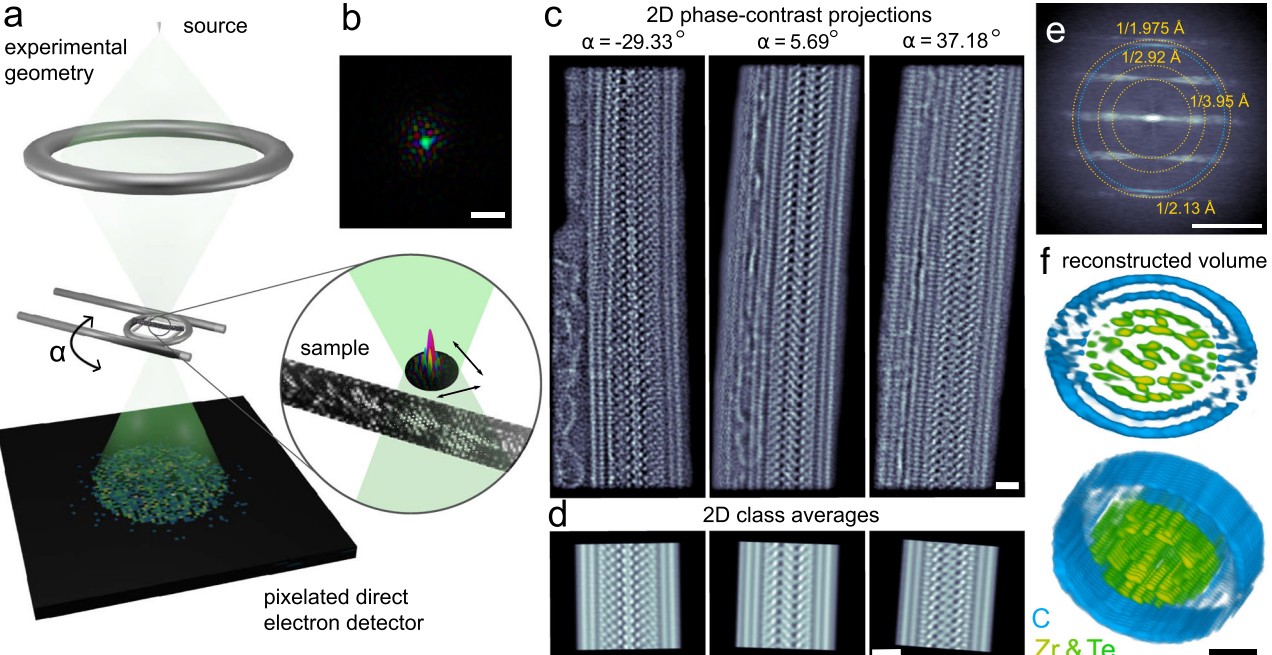

**Fig. 1 | Ptychographic atomic electron tomography (PAET) of a complex nanostructure. a** A converged electron beam is scanned over a nanoscale sample which is tilted around the $\alpha$ axis of a high-precision tomography holder. Four-dimensional scanning diffraction datasets are recorded at every tilt angle using a direct electron detector. **b** Reconstructed complex wave function of an example STEM probe. **c** Examples of phase-contrast projection images, reconstructed using mixed-state electron ptychography. **d** High-SNR class averages calculated from the images in (**c**). **e** Average diffraction pattern of all 36 tilt angles with some scattering angles labeled. **f** Two orientations of the 3D reconstructed volume from the 2D unit cell averages in (**d**). All real space scales bars are 10 Å, scale bar in (**e**) is 0.5 Å$^{-1}$.

variation in the structural sub-units making up each. Most atoms in the Zr-Te core and shell can be directly resolved, while some atoms close to the edges along the missing wedge direction appear as elongated columns. From the reconstructed volume, we have measured the 3D positions of 580 Zr and Te atoms to subpixel precision using the procedures developed in AET routines[6,14]. The chemical species of the Te and Zr atoms were determined by analyzing the number of nearest neighbors and the local coordination (Methods section G).

The configuration and elliptical dimensions of the DW-CNT were determined by first excluding chiral CNTs based on the Fourier spectrum of the projections shown in Fig. 1e. The armchair configuration was excluded based on the observation of a strong reflection at 1/2.13 Å$^{-1}$ in the direction of the nanotube, which is present only in zig-zag nanotubes where the 2.13 Å spacing is oriented across the nanotubes. The dimensions of the CNTs were determined by maximizing the agreement of the atomic coordinates with the reconstructed projected intensity of the CNTs from the volume in Fig. 1f. We determined best-fit chiral indices of (50, 0) and (39, 0) for the outer and inner CNT.

## Atomic structure

Figure 2a shows the full atomic model overlaid over a 2D slice through the reconstructed phase-contrast volume. Fig. 2b shows a three-dimensional rendering of the structure, with the outermost Zr-Te shells peeled back to display the local coordination. Inside the DW-CNT, eight one-dimensional $ZrTe_5$ clusters encapsulate a three unit cell wide $ZrTe_2$ structure. The outer $ZrTe_5$ clusters are split into two groups of 4 $ZrTe_5$ sub-units, shown in Fig. 2c. Each cluster has a 1D chain structure along the nanotube axis, with the same orientations[58]. The four groups are circumscribed inside the inner CNT wall, with an angle of ~26° between each cluster and a radius of 14 Å to the central Zr atom. The overall geometric arrangement of the full structure is shown in Fig. 2d, e, with the DW-CNT major and minor axes lengths labeled.

On the top and bottom of the encapsulated Zr-Te structure, there are two single-atom Te chains. One-dimensional Te-Te chains have

been observed before as stable structures encapsulated in CNTs[59]. The relative position of the Te-Te chains and the $ZrTe_2$ core structure are shown in Fig. 2f. The $ZrTe_2$ is a previously unobserved phase of Zr-Te, which we examine in more detail in the following section. The overall structure has a stoichiometry of $Zr_{11}Te_{50}$, and a striking elliptical structure which may be caused by the intercalation of the ZrTe structures into the DW-CNT.

## Electronic structure

To investigate the stability and electronic structure of our proposed $Zr_{11}Te_{50}$ structure, we have performed first principles calculations, as described in Methods Sec. J. The resulting optimized atomic structure is depicted in Fig. 3a. As previously discussed, the outer tube is well described as one-dimensional chains of $ZrTe_5$ which consist of face-sharing Zr-Te polyhedra where each Zr is 8-coordinated with Te. These 8-coordinated Zr-Te chains are distorted versions of those that make up bulk $Cmcm$ $ZrTe_5$ crystal where each Zr has three pairs of identical Zr-Te bond lengths, with the two remaining Zr-Te bonds being different. The coordination of the outer ring of Zr in our $Zr_{11}Te_{50}$ is structurally the same with slight deviations from the "paired" Zr-Te bond lengths (e.g., 2.955 Å -vs- 2.960 Å). We present the calculated electronic band structure of the bulk $ZrTe_5$ from which our outer structure is derived in Fig. 3d—we confirm the previously-reported topological insulating phase with the band inversion near to $\Gamma$[60].

The inner core $ZrTe_2$ structure presents an intriguing case for structural analysis. We find two different coordinations for Zr in the innermost layer. The central Zr is 8-fold coordinated with Te, with the other two Zr sites being 6-fold coordinated with Te. We next isolated this $ZrTe_2$ structural unit and performed a full structural optimization to remove the influence of any confinement/pressure from the sandwich structure, with the result depicted in Fig. 3c. This new $ZrTe_2$ structure adopts the space group $Pmmm$, and forms a thin two-dimensional slab. In fact, the ground state polymorph, $1T$-$ZrTe_2$ (space group $P\bar{3}m1$), has only recently been grown by molecular beam

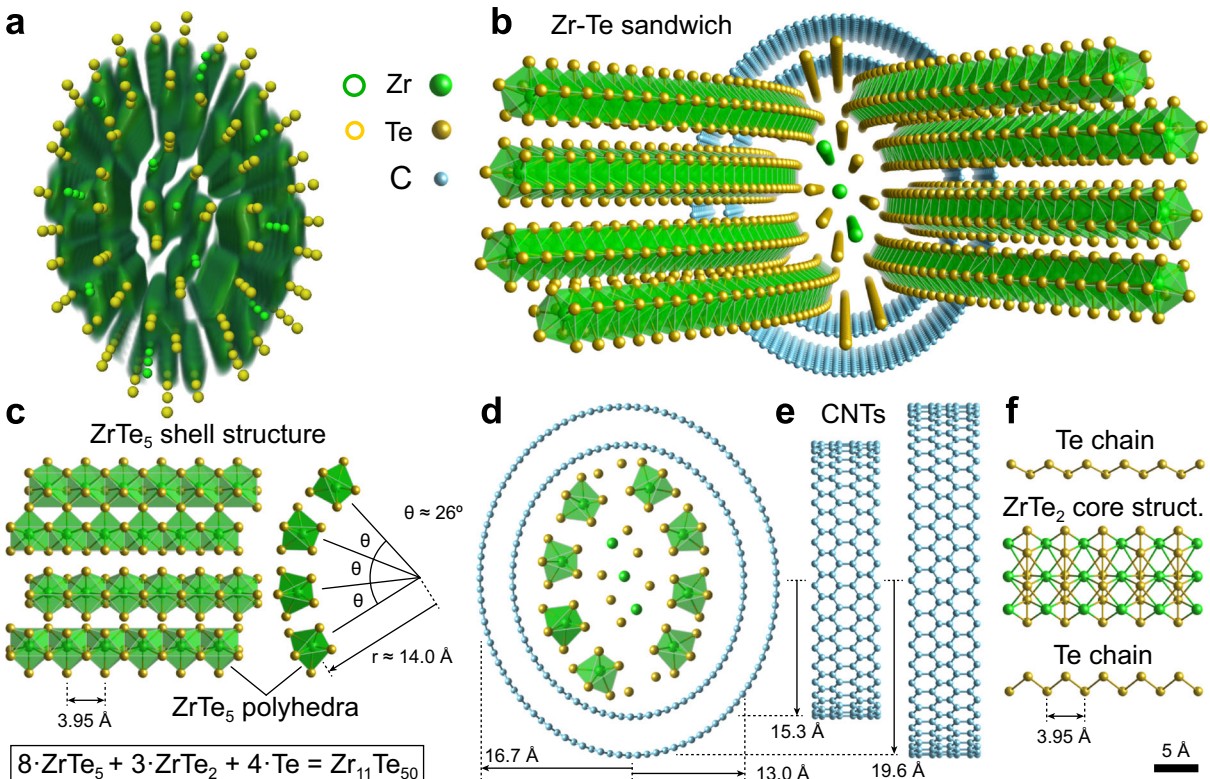

**Fig. 2 | DW-CNT-encapsulated $Zr_{11}Te_{50}$ sandwich structure. a** atomic model overlaid over 3D reconstructed volume of the core structure. **b** Three-dimensional model of the $Zr_{11}Te_{50}$ structure with folded-out one-dimensional $ZrTe_5$ chains. **c** Left panel: side view of four $ZrTe_5$ chains building one side of the sandwich structure. Right panel: Front view of four $ZrTe_5$ chains of one side of the sandwich structure. **d** Front view of the full atomic model showing the elliptical CNTs and the spatial extents of their semi-minor and semi-major axes. **e** Side view of the DW-CNT with zig-zag configuration. **f** Side view of the $ZrTe_2$ core structure and the Te chains.

epitaxy[61]. This $1T$ phase is 6-fold coordinated and forms a van-der-Waals two-dimensional structure with measurements and theory suggesting it is a Dirac semimetal[61]. The innermost Zr coordination is more unusual. While this Zr site is 8-fold coordinated as in the $ZrTe_5$ chains, it now forms a regular octahedron where each Zr-Te bond length is the same. The calculated electronic band structure of this new structural phase is shown in Fig. 3e. Our $Pmmm$ $ZrTe_2$ phase is a metal with primarily hole pockets at the Fermi level – the low-dimensional nature of the structure is also confirmed with the flat bands in the X to S directions (short axis). Finally, using symmetry indicators, we predict this new phase to be a Dirac semimetal, akin to its stable $P\bar{3}m1$ polymorph[61].

## Discussion

In summary, we have experimentally demonstrated atomic structure determination of a complex nanomaterial by phase-contrast ptychographic AET combined with unit cell averaging and DFT calculations. We have determined the chiral numbers of the encapsulating DW-CNT from the 2D projections and determined its ellipticity from the reconstructed volume. In addition to showing contrast improvement compared to AET for the weakly-scattering carbon atoms, PAET simultaneously recovers partial coherence effects, probe positions, and probe aberrations present in the experiment. This, in turn, allows focusing at lower resolution and an overall lower dose due to less pre-exposure irradiation of the sample. Further improvements in the experimental protocol will allow skipping the unit cell averaging step in the reconstruction and enable imaging of single light atoms at atomic resolution. The reconstructed volume achieves a resolution of 1.02 Å and 1.1 Å for the Zr and Te atoms, respectively, perpendicular to the missing wedge direction and 2.26 Å along the missing wedge direction (Supplementary Figs. 16, 17). We expect that 3D phase-

contrast reconstructions with sub-Ångström resolution will soon be possible with further advancements in detectors, reconstruction algorithms, and experimental protocols. We have used PAET to solve the structure of a DW-CNT-encapsulated complex $Zr_{11}Te_{50}$ nanowire structure, which contains both previously observed linear chain structures of $ZrTe_5$ and Te-Te, and a previously unobserved $ZrTe_2$ structure. Density functional calculations both confirm the stability of our Zr-Te model structure and elucidate its electronic properties. We expect that PAET will find widespread application for solving the structures of complex materials that contain light elements, weakly-scattering structures, and beam-sensitive materials.

## Methods

### Sample preparation

Encapsulated $Zr_{11}Te_{50}$ species are synthesized within CNTs using protocols similar to those for the growth of confined $TaTe_2$, $NbSe_3$, and $HfTe_3$ structures[51,52,57]. The $Zr_{11}Te_{50}$ species can be synthesized following the synthesis of $ZrTe_3$, where stoichiometric quantities of Zr powder and Te shot (450 mg total), 1mg to 2mg of end-opened multi-walled CNTs with an inner diameter ranging from 1.0 nm to 10.0 nm (CheapTubes, 90% SW-DW-CNT), and 5 mg/cm³(ampoule volume) of $I_2$ are sealed under vacuum ($1.33 \times 10^{-6}$ mbar) in a quartz ampoule. The ampoule is heated in a single-zone furnace at 550° for 3–5 days, cooled to 350° over 3 days, then cooled to room temperature over 1–2 days. The Zr-Te filled CNTs are dispersed in isopropyl alcohol by bath sonication for 1 h and drop-cast onto lacey carbon TEM grids for subsequent electron microscopy analysis.

### Data acquisition

A tomographic tilt series was acquired from a $Zr_{11}Te_{50}$ DW-CNT using the TEAM 0.5 microscope and TEAM stage[62] at the National

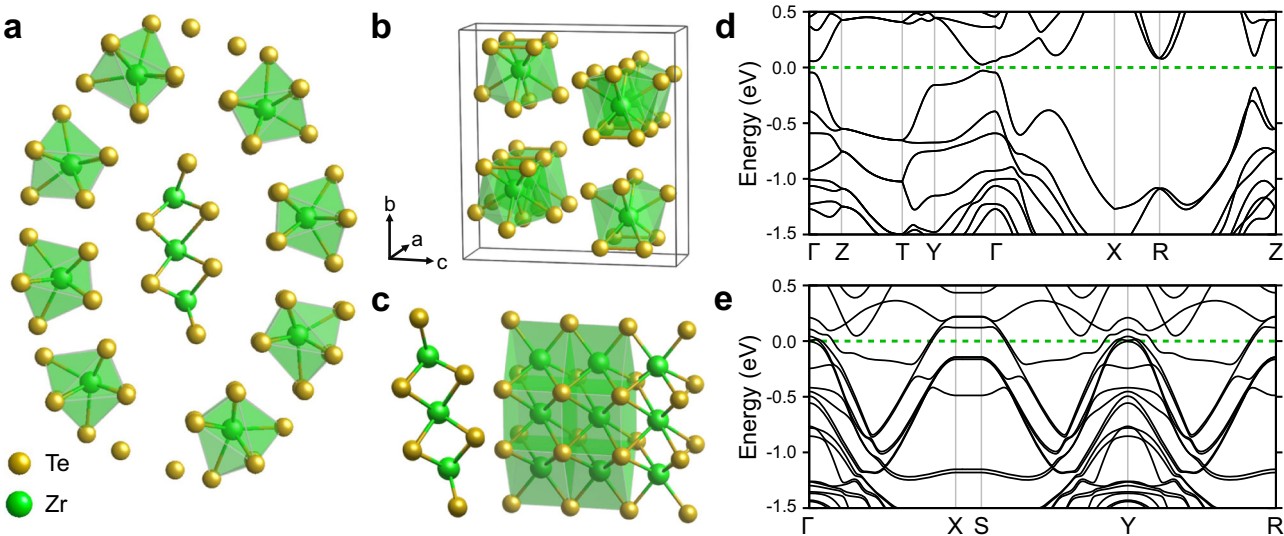

**Fig. 3 | First principles calculations of $Zr_{11}Te_{50}$. a** DFT optimized $Zr_{11}Te_{50}$ structure. **b** Crystal structure of *Cmcm* $ZrTe_5$ (**c**) Fully optimized inner $ZrTe_2$ structural unit (**d**) Calculated electronic band structure of *Cmcm* $ZrTe_5$. **e** Calculated electronic band structure of inner $ZrTe_2$ structural unit. For both band structures, the Fermi level is set to 0 eV and is marked by the green dashed line.

Center for Electron Microscopy in the Molecular Foundry. Before the tilt series, the TEM grid square (300 mesh) which contained the sample was beam showered for 30 min with 2 nA current, giving a pre-exposure fluence of $2\,e/Å^2$. We recorded four dimensional-scanning transmission electron microscopy (4D-STEM) datasets[63] with full diffraction patterns over $1600 \times 600$ probe positions at each tilt angle. The diffraction pattern images were acquired with the 4D Camera prototype, in-house developed in collaboration with Gatan Inc., a direct electron detector with $576 \times 576$ pixels and a frame rate of 87 kHz[64], at 80 kV in STEM mode with a 25 mrad convergence semi-angle, a beam current of 40 pA, estimated from 4D camera counts, a real-space pixel size of 0.4 Å, and camera reciprocal space sampling of 173.6 µrad per pixel. These settings amounted to an accumulated fluence of $1.79 \times 10^4\,e/Å^2$ per projection and $6.28 \times 10^5\,e/Å^2$ for the whole tilt series. The tilt series was collected at 38 angles with a tilt range of +63 to −58°. To minimize the total electron exposure, focusing was performed at a resolution of 80 kX before switching to high magnification for data collection.

**Ptychographic reconstruction**
The raw 4D-STEM datasets of size 650 GB per tilt angle and 22.75 TB in total were electron-counted using the open-source stempy software[65] on the Cori supercomputer at NERSC and saved in a sparse linear-index encoded electron event representation (EER)[66], using 6.5 GB storage per tilt. Crop-outs containing only the scan positions covering the $Zr_{11}Te_{50}$ DW-CNT for all tilts were determined from annular bright-field images (12.5 mrad–25 mrad integration range) of the tilt series for ptychographic reconstruction. Initial guesses for the defocus aberration for each tilt were manually obtained with an interactive real-time implementation[28] of the single-sideband ptychography method[67]. The data in EER format was further preprocessed by centering the position-averaged diffraction pattern, symmetric cropping to a maximum detector angle of $2\alpha$, with $\alpha$ the semi-convergence angle, and binning to a detector size of $88 \times 88$ pixels. The final cropped and preprocessed data of the whole tilt series, compressed with the gzip compression algorithm, had a total size of 12.3 GB. Phase-contrast images, probe positions and a low-rank approximation of the partially coherent illumination were jointly reconstructed using 115 iterations of the Least-Squares Maximum Likelihood (LSQML) method with gradient-based scan

position correction[33,68], with the parameters in Supplementary Table II. The maximum thickness for a sample to fulfill the projection approximation of ptychography for a numerical aperture of 25 mrad is 9 nm. The largest dimension of the DW-CNT along the beam direction is 3.9 nm, it is therefore within the limit of the projection approximation for single-slice ptychography. Due to the slight tilt of the nanotube, the projected thickness reaches 6 nm for some projections. For the best reconstruction quality of projections, we consecutively reconstruct chunks of the nanotube, and feed the probe from the last reconstructed chunk in the reconstruction of the next chunk. The chunking scheme is shown in Supplementary Fig. 11. We note that this chunking scheme slightly improved the visual quality of the reconstructions. Reconstruction without this scheme is also possible, see for example, Supplementary Fig. 2 for a full field of view reconstruction.

**Preprocessing for phase-contrast tomography**
Since low spatial frequencies are weakly transferred in noisy 4D-STEM datasets collected close to the focus condition[69], the ptychographic tilt series reconstructions display characteristic halos around the nanotube. With the prior knowledge that the nanotubes are suspended over vacuum, we manually create vacuum masks around the nanowire and set the vacuum phase equal to the lowest phase value of the halo close to the nanotube. We then performed a first alignment using the local center-of-mass method[70].

**Tomographic reconstruction**
From the initially-aligned tilt series, a 3D reconstruction was performed using joint reconstruction and rigid alignment as in[71], implemented with the automatic gradient calculation of the pytorch package[72]. First, coarse alignment was performed at a downsampled projection size to 12% with 4000 iterations, followed by a second round of 5000 iterations at full resolution. The 2D class averaged projections were then aligned with cross-correlation to the final aligned full projections. Using this approach, the reconstruction with the 2D class averages converged to a minimum R-factor of 5.4%. The reconstructed volume from the full 2PD projections is $27.7 \times 11.6 \times 11.6$ nm in size, while the volume reconstructed from 2D class averages is $5.88 \times 5.3 \times 5.3$ nm in size. We note that the total thickness of the DW-CNT, even including bending, is smaller than the

depth of field. (See Supplementary Fig. 10). Tomviz[73] was used for visualization of the reconstructed volume.

## Atom tracing

The 3D atomic positions of the Zr and Te atoms were determined using the following procedure developed in[10]. First, All local intensity maxima were identified from the final 3D reconstruction. Starting from the highest intensity peak, a 3D Gaussian function of $7 \times 7 \times 7$ voxels was fitted to the peak. Second, if this peak was satisfied with a minimum distance constraint (that is, the distance between two neighboring atoms was larger than 3 Å), it was added to a peak position list. The rotation of the nanotube with respect to the beam propagation normal plane was determined as 5.2° and the candidate atoms were then rotated onto this normal plane. Atomic columns along the tube were identified by projecting the volume along the tube and 2D peak finding. The x-y coordinates of atom candidates that were farther than 2 Å from the next column were set to the value of the intensity peak of the projection along the tube, and duplicate atoms were removed. Some missed atoms were added by taking a line trace along the tube axis at positions of projection maxima and 1D peak finding along this trace. This procedure resulted in 580 candidate atoms.

## Atomic model construction

From the candidate atoms, we have constructed the Zr-Te model structure. We note that for most sites, it was not possible to directly determine the atomic species from the reconstructed 3D phase-contrast signal at the experimental signal-to-noise ratio. The prototype 4D camera detector is mounted off-center relative to the ADF detectors, such that the simultaneous ADF signal could not be used for tomographic reconstruction. We expect better chemical contrast in future experiments with both ADF and 4D-STEM signals[12,74]. The development of joint reconstruction methods for such multi-modal tomography datasets is a promising avenue for future research. Chemical species of all but two atomic columns in the nanotube could be determined from the coordination environment with the following procedure. For each atom candidate, we determined the number of nearest neighbors (NNs) in a shell of 3.1 Å radius. As candidate stable structures we considered all known stable Zr and Te containing structures, including $ZrTe_2$, $ZrTe_3$, and $ZrTe_5$. In those structures, the Zr atoms are bonded to six or eight Te atoms, while the Te atoms can be bonded in different geometries to 2–4 Zr atoms. We therefore created two classes for atoms with less than four neighbors and those with more than four neighbors. The result of this classification is shown in Supplementary Fig 7. We then assigned the columns with majority of atoms with more than 4 NNs as Zr atoms and the columns with majority of atoms in the class with less than 4 NNs as Te atoms.

This procedure left two atomic columns in the center of the nanotube that could not be assigned based on NNs. These are marked with circles in Supplementary Fig 7. Those two ambiguous site identities were tested using DFT calculations, where in all cases we used the lower energy stoichiometries. From the chemical classification based on NNs and the candidate atomic positions, the eight outermost $ZrTe_5$ units and their positions/angles were identified. Next, we identified two isolated chains of atoms following a zig-zag pattern at the top and bottom of the tube as positioned in Fig. 2. These sites were assumed to be composed of Te atoms, due to previous observations of Te chains encapsulated in carbon nanotubes[59], and the presence of weakly-bound Te chains in the $ZrTe_5$ structural units. Finally, the core was determined to be composed of a central Zr atom with 8 Te neighbors and two partially-coordinated Zr atoms, forming a small 2D ribbon of $ZrTe_2$ with a previously unreported structure. The final atomic model and reconstruction volume are illustrated in Supplementary Movie 1.

The DW-CNT configuration was determined in the following way. CNTs are characterized by their chiral indices (n,m). Chiral nanotubes can be excluded for our sample because the diffraction pattern of chiral nanotubes follows a Bessel function radial symmetry[75] with oscillating layer lines. In the diffraction patterns of the experimental projections in Supplementary Fig. 1 (insets) it can be seen that the 1/2.13 Å$^{-1}$ graphene reflection is not an oscillating layer line. This leaves armchair configuration (k,k) or zig-zag configuration (k,0) nanotubes as possible options. In armchair configuration nanotubes, the 2.13 Å spacing lies across the nanotube and can be suppressed by material in the tube, while in zig-zag configuration the 2.13 Å spacing lies along the tube and is strongly visible at the edges of the tube where it is overlapped by no other material. This is the case in our sample, and we, therefore, exclude the armchair configuration, which leaves us to determine the chiral indices $k_1$ and $k_2$ of the outer and inner nanotube, and the determination of the semi-minor and semi-major axis dimensions of the nanotube. We determined the chiral indices by maximizing the the intensity of the pixels sampled by CNT atomic coordinates overlaid over the 2D projection of the reconstructed volume along the tube, as shown in Supplementary Fig 7. The sampled intensity was maximized for chiral numbers $k_1=50$ and $k_2=39$, with semi-minor axes of 16.7 Å and 13.0 Å and semi-major axes of 19.6 Å and 15.3 Å. The only parameter that could not be determined uniquely was the relative rotation of the CNTs to each other, as the 2D unit cell averaging procedure reduced the atomic contrast for the CNT. As such, the relative rotation of the CNTs shown in Fig. 2 is only one possible configuration.

The Analyses in this study were performed using the Python and Matlab programming languages. Portions of Fig. 1 were generated using Matplotlib[76], and portions of Figs. 1, 2, and 3 were plotted using the VESTA program[77].

## Estimation of precision

To calculate the precision of our position measurements, we first investigated if the nanotube can be modeled with a linear model or if a full multislice treatment is necessary. To this end, we simulated one projection both with a linear forward model as in ref. [10] and with a multislice simulation incorporating all partial spatial and temporal coherence effects, followed by mixed-state ptychographic reconstruction and unit cell averaging. First, we created 12 different atomic models where the Zr and Te atoms and atom positions are identical, but the carbon nanotube is displaced by 2.13/12 Å every time, such that the averaging removes atomic contrast of the nanotube. We then fitted a linear model to this average structure by determining the H- and B-factors that minimize the R-factor between model and experimental reconstruction. This linear model is shown in the middle panel of Supplementary Fig. 4 and achieves and R-Factor of 12%. Then, we performed multislice 4D-STEM simulations using the PRISM algorithm[78] in the abTEM simulation code[79], incorporating partial spatial coherence with a FWHM source size of 80 pm and partial temporal coherence with a chromatic defocus spread of 6 nm, as calculated from the chromatic aberration coefficient and energy spread of the TEAM 0.5 microscope at 80 keV, and with the experimental dose of $1.72 \times 10^4$ $e$/Å$^2$ per projection.

We then performed mixed-state ptychographic reconstruction using the parameters in Supplementary Table II, and averaged the resulting phase-contrast images. The resulting 2D class average is shown in the right panel of Supplementary Fig. 4. We see that the ptychographic reconstruction from full-multislice simulated data is overall sharper than the experimental reconstruction. The overall R-factor between experiment and mixed-state ptychographic reconstruction from multi-slice simulation is 16%, while for the linear model it is 15%. Possible reasons for the better quality of the reconstruction from simulated data, even including all partial coherence and dose effects, are imperfect detector quantum efficiency and modulation transfer function of the camera, which was not modeled in the simulations, and slight differences in the structure along the tube. Especially amorphous carbon wrapped around the tube, which was very

helpful for alignment of the projections, causes resolution loss compared to the simulated structure.

Because of the better agreement of the linear model with the experiment, and the much lower computational complexity required for simulation, we performed subsequent tomographic reconstructions from model-generated projections with the linear model. We generate 28 projections with the linear forward model as above and reconstruct the volume with the same algorithm and alignment strategy as the experimental dataset, described in Section E. We then trace the atomic coordinates in the reconstructed volume with the same parameters as the experimental projections as described in Section F, resulting in 580 traced atomic coordinates. The traced coordinates are then aligned to the experimental coordinates with the iterative closest point algorithm, and atoms obtained from simulation paired with the nearest traced atom from experiment. We then determine the displacement between experimental and simulated, retraced atoms. supplementary Fig. 6a shows a histogram of the RMS displacement of all 580 paired atoms, with a mean position error of 17 pm and median position error of 10 pm. Supplementary Fig. 6b shows the spatial distribution of the displacement errors. It can be seen that Zr & Te atoms close to the top and bottom of the nanotube, where the missing wedge artefacts are strongest, have the highest position error.

### Resolution

We estimate the resolution similar to ref. [46]. First, we extract the mean Zr and mean Te atomic volumes from the experimental reconstruction and upsample to 0.1 Å pixel size. Then we simulate Zr and Te atomic potentials with the abTEM[79] package by averaging 250 frozen phonons at 293 K, with standard deviation $\sigma = 0.085$ Å calculated from crystalline Zr[80] and convert them to the 3D transmission function. Subsequently, we solve an optimization problem that minimizes the difference between a Gaussian PSF-blurred transmission function and the mean experimental atom volume for Zr and Te atoms. The FWHM of the determined PSF follows approximately the Abbe resolution[46]. For the mean Zr atom, we determine a resolution of $d_\perp = 1.02$ Å perpendicular to the missing wedge and $d_\parallel = 2.26$ Å along the missing wedge direction, giving a mean 3D resolution of $d_{Zr} = 1.63$ Å. For the mean Te atom we determine a resolution of $d_\perp = 1.1$ Å perpendicular to the missing wedge and $d_\parallel = 2.26$ Å along the missing wedge direction, giving a mean 3D resolution of $d_{Te} = 1.68$ Å. The mean experimental, simulated, and optimized PSF-blurred volumes for Zr and Te are shown in Supplementary Figs. 16, 17, respectively.

### Density functional theory calculations

Previous first principles calculations on $ZrTe_5$ have emphasized the importance of accurately treating dispersive interactions to reproduce experimental structural parameters, with a comparison given in the SI, and the resulting electronic and topological properties[81]. Because of this, we first performed structural optimizations on bulk $ZrTe_5$ to benchmark our choice of exchange-correlation functional with dispersive corrections. We find that the optB86b-vdW functional gives lattice parameters that are less than 1% different than those measured by powder diffraction, a significant improvement over PBE, which gives an error of over 9% for the *b* lattice parameter. (See further details in the SI).

Owing to the prohibitively large unit cell size of the full Zr-Te/CNT structure depicted in Fig. 2b which comprises over 5000 atoms for full ab initio treatment, we instead perform calculations of the inner Zr-Te sandwich structure. To simulate the confining influence of the DW-CNT around the Zr-Te sandwich, we fix the outermost Te atoms during the structural optimization, while allowing the remaining Zr and Te positions to relax. We perform fixed-volume calculations for several values of the short *a* lattice parameter, finding a minimum at 3.97 Å, very close to the value of 3.95 Å suggested from our diffraction analysis. As a test of the stability of the $Zr_{11}Te_{50}$ structure without the confining influence of the CNTs, we also performed a fixed-volume

structural optimization allowing all atomic positions to relax *without* including van-der-Waals corrections (i.e., using the PBE exchange correlation functional). With PBE only, we find the $Zr_{11}Te_{50}$ structure expands, which is primarily due to the separation between the structural units increasing and not changes in bond lengths or configurations, as we would expect without the dispersive corrections.

## Data availability

The raw data is available at Zenodo under the accession code https://doi.org/10.5281/zenodo.8300707 in hdf5 format.

## Code availability

The code and resulting datasets are available at Zenodo under the accession code https://doi.org/10.5281/zenodo.8380800.

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

## Acknowledgements

P.M.P. is supported by the Strobe STC research center, Grant No. DMR 1548924 and by an EAM starting grant. S.M.G. was supported by the Quantum Systems Accelerator (QSA) funded by the Department of Energy, Office of Science, National Quantum Information Science Centers. H.D.V. is supported by NSF-DMR award number 1848079. M.C.S. is supported by the Strobe STC research center, Grant No. DMR 1548924. C.O. acknowledges support from the Department of Energy Early Career Research Award program. Work at the Molecular Foundry was supported by the Office of Science, Office of Basic Energy Sciences, of the U.S. Department of Energy under Contract No. DE-AC02-05CH11231. S.S., D.P., and A.Z. acknowledge support by the U.S. Department of Energy, Office of Science, Office of Basic Energy Sciences, Materials Sciences and Engineering Division (DE-AC02-05-CH11231), primarily within the van der Waals Heterostructures (KCWF16) Program, which provided for synthesis, and within the Nanomachine program (KC1203) which provided for high resolution TEM. This research used resources of the National Energy Research Scientific Computing Center (NERSC), a U.S. Department of Energy Office of Science User Facility located at Lawrence Berkeley National Laboratory, operated under Contract No. DE-AC02-05CH11231 using NERSC awards ERCAP0020898, ERCAP0020897 and ERCAP0024699. P.M.P. gratefully acknowledges the scientific support and HPC resources provided by the Erlangen National High Performance Computing Center (NHR@FAU) of the Friedrich-Alexander-Universität Erlangen-Nürnberg (FAU) under the NHR project AtomicTomo3D. NHR funding is provided by federal and Bavarian state authorities. NHR@FAU hardware is partially funded by the German Research Foundation (DFG) - 440719683. The 4D Camera was developed under the DOE BES Accelerator and Detector Research Program, with collaboration from Gatan, Inc.We would like to thank Gatan, Inc. as well as P. Denes, A. Minor, J. Ciston, J. Joseph, and I. Johnson who contributed to the development of the 4D Camera.

## Author contributions

S.S. and D.P. prepared and screened CNT-encapsulated Zr-Te samples under supervision of A.Z. P.E. developed parts of the 4D Camera control and data management software. P.M.P. carried out the 4D-STEM experiments, assisted by P.E. M.C.S. performed tilt axis alignment measurements. P.M.P. implemented the data preprocessing codes and adapted the mixed-state ptychography reconstruction for 4D Camera data. P.M.P. and H.D.V. performed mixed-state ptychography reconstructions. P.M.P. developed the tomographic reconstruction codes and performed tomographic reconstructions. C.O. developed elliptical tube fitting codes, template matching, and unit cell extraction codes. P.M.P. and C.O. performed unit cell averaging and atom tracing. P.M.P. performed multi-slice simulations and precision estimation. C.O., P.M.P., and D.P. discussed and solved the atomic structure. S.M.G. performed DFT calculations, using input models created by C.O. P.M.P. conceived the study under discussion with M.C.S, C.O., and S.S. P.M.P., C.O., M.C.S., and S.M.G. wrote the manuscript with input from all authors.

## Competing interests

The authors declare no competing interests.
