## [Peer Review File · Nature Communications]

Solving Complex Nanostructures With Ptychographic Atomic Electron TomographyEditorial Note: This manuscript has been previously reviewed at another journal that is not operating a transparent peer review scheme. This document only contains reviewer comments and rebuttal letters for versions considered at *Nature Communications* .

REVIEWER COMMENTS

Reviewer #1 (Remarks to the Author):

I thank the authors for their responses to my comments on the initial submission. While I still have some concerns that the approach taken here is quite intricate and is probably very specific to the materials system being studied, I agree with one of the other referees that this is an heroic experiment and I am keen to see it published.

In answer to my question about whether ADF-based atomic-resolution tomography would have been more appropriate for the Zn and Te species, the authors respond that ptychography offers better signal to noise and allows for focus variation correction. These are both valid arguments. The signal to noise issue is highlighted several times, but perhaps the authors might like to mention the defocus correction in any further changes they might make at this stage in the main manuscript. This strikes me as an important advantage over incoherent tomography modes such as ADF in which changes in sample height must be corrected with a focus change.

There is much else that could be discussed, but the paper is scientifically sound and I don't wish to hold up publication. I recommend publication.

Reviewer #4 (Remarks to the Author):

In this manuscript, the authors have demonstrated that the 3D atomic structure of compounds containing both light and heavy elements can be determined using ptychographic atomic electron tomography (PAET). The authors have done an excellent job here, considering that collecting and processing datasets of this kind is not straightforward. One advantage of using ptychography in PAET method over annular dark-field imaging (ADF) in conventional atomic electron tomography technique in STEM is the ability to determine the positions of both light and heavy atoms in compounds containing light and heavy elements (e.g. light carbon atoms and heavy Zr and Te atoms in this work). However, determining the type of resolved atoms in the phase images reconstructed using ptychography is not straight forward.

In this work, the authors resolved the position of C, Zr and Te atoms in a double-wall nanotube encapsulating a complex ZrTe structure using PAET. They claim that the encapsulated phase is a complex $Zr_{11}Te_{50}$ containing $ZrTe_2$ (core), $ZrTe_5$ (shell) and Te-Te chains. The authors have performed a nice work to reveal the structure of a complex phase encapsulated in a double-wall nanotube. Below are certain concerns arise regarding the accuracy of the presented results. If the authors can address the following concerns, this manuscript is recommended for publication in Nature Communications.

a) Atom position determination:

A nanotube is not entirely free of bends along its axes; that is, certain sections of the nanotube may deviate by a few degrees from being perpendicular to the beam direction in TEM. As a result, there can be a considerable variation in height from one side of the nanotube to the other, i.e. a significant defocus difference along the length of the nanotube. Furthermore, if the nanotube axis is not perpendicular to the tilting axis, such a defocus difference will be further amplified when tilting is involved.

In this work, the phase reconstruction has been performed in a large field of view (e.g. it is ~ 20 nm

along the nanotube axis for Fig. 1 and ~ 50 nm for Fig. S2) and large tilting range (-58 to +63 degrees). The defocus difference along the nanotube in such experimental conditions can easily be quite larger than the depth of focus. If the nanotube examined in this work was not entirely free of bends and its axis was not almost perpendicular to the rotation axis, the defocus difference along the nanotube would be quite larger than the depth of focus (e.g. quite larger than 10 nm in some tilting angles). Such a large defocus difference across the image affects the accuracy of the atom position determination in the reconstructed ptychographic phase images. The authors dealt with this problem by dividing the 4D datasets in few slices along the nanotube and performed reconstruction for each slice separately as noted in the manuscript: "For the best reconstruction quality of projections where the nanotube is tilted along the beam direction, we reconstruct slices along the nanotube separately with different probes". The authors should provide further information regarding this procedure to demonstrate that the defocus difference along the nanotube has been carefully considered in the phase reconstruction. For example what was the size of those slices and how the sudden change in the defocus between slices affects the accuracy of the reconstructed phase?

In addition, the authors provided a single probe function for each dataset in Fig. 1 as well as Fig. S5 while they used multiple probes for some (or all) of their datasets. It is not clear that those probes are for the whole or a part of those datasets. Furthermore, could the authors explain the considerable change in the reconstructed probe function caused by tilting observed in Fig. S5? Does those changes related solely to the defocus variation? (which I don't think is the case) It seems that the other aberrations are changing to compensate the change in the defocus variation across the field of view. If that is the case, it can considerably affect the accuracy of the results.

b) Identification of atoms

Although ptychography is a powerful technique to obtain information about the position of atoms in a structure, it is not a proper technique for identification of the type of atoms in a sample. Hence, the sample should be examined by some quantitative techniques (e.g. ADF) to support the atomic structure determined in this work.

The authors provided a comparison of ADF vs iCOM vs ptychography (Fig. S2) at the electron dose which they used for their tomography experiment to demonstrate that ADF cannot be used to obtain useful information from the sample due to the low the signal-to-noise ratio in their experimental condition. However, the electron dose used in their experiment has been chosen for the sample to tolerate electron irradiation during several times of data collection from one area. I believe that it is possible to use much higher electron dose to collect ADF images from this sample which can be used to discriminate between Zn and Te atoms. Alternatively, they can obtain ADF images with a suitable signal-to-noise ratio by averaging over >30 ADF images from a single area using the same experimental conditions employed in this work.

Finally, it would be great if the authors could provide information on the best achieved resolution (2D and 3D) in the manuscript.

Response to the Reviewers

(Dated: July 19, 2023)

We thank all reviewers for their detailed feedback. Below are our point-by-point responses to the remaining comments and questions. Thank you for your time!

Reviewer 1 (Remarks to the Author):

I thank the authors for their responses to my comments on the initial submission. While I still have some concerns that the approach taken here is quite intricate and is probably very specific to the materials system being studied, I agree with one of the other referees that this is an heroic experiment and I am keen to see it published. In answer to my question about whether ADF-based atomic-resolution tomography would have been more appropriate for the Zn and Te species, the authors respond that ptychography offers better signal to noise and allows for focus variation correction. These are both valid arguments. The signal to noise issue is highlighted several times, but perhaps the authors might like to mention the defocus correction in any further changes they might make at this stage in the main manuscript. This strikes me as an important advantage over incoherent tomography modes such as ADF in which changes in sample height must be corrected with a focus change. There is much else that could be discussed, but the paper is scientifically sound and I don't wish to hold up publication. I recommend publication.

Thank you for the positive comments! We have added the following sentence mentioning explicitly the beneficial feature of joint reconstruction of the probe.

This feature of modern ptychographic reconstruction algorithms is especially beneficial to reduce the total dose in atomic-resolution experiments because precise focusing before the acquisition is not necessary using PAET.

Reviewer 4 (Remarks to the Author):

In this manuscript, the authors have demonstrated that the 3D atomic structure of compounds containing both light and heavy elements can be determined using ptychographic atomic electron tomography (PAET). The authors have done an excellent job here, considering that collecting and processing datasets of this kind is not straightforward. One advantage of using ptychography in PAET method over annular dark-field imaging (ADF) in conventional atomic electron tomography technique in STEM is the ability to determine the positions of both light and heavy atoms in compounds containing light and heavy elements (e.g. light carbon atoms and heavy Zr and Te atoms in this work). However, determining the type of resolved atoms in the phase images reconstructed using ptychography is not straight forward.

In this work, the authors resolved the position of C, Zr and Te atoms in a double-wall nanotube encapsulating a complex ZrTe structure using PAET. They claim that the encapsulated phase is a complex $Zr_{11}Te_{50}$ containing $ZrTe_2$ (core), $ZrTe_5$ (shell) and Te-Te chains. The authors have performed a nice work to reveal the structure of a complex phase encapsulated in a double-wall nanotube. Below are certain concerns arise regarding the accuracy of the presented results. If the authors can address the following concerns, this manuscript is recommended for publication in Nature Communications.

a) Atom position determination: A nanotube is not entirely free of bends along its axes; that is, certain sections of the nanotube may deviate by a few degrees from being perpendicular to the beam direction in TEM. As a result, there can be a considerable variation in height from one side of the nanotube to the other, i.e. a significant defocus difference along the length of the nanotube. Furthermore, if the nanotube axis is not perpendicular to the tilting axis, such a defocus difference will be further amplified when tilting is involved. In this work, the phase reconstruction has been performed in a large field of view (e.g. it is 20 nm along the nanotube axis for Fig. 1 and 50 nm for Fig. S2) and large tilting range (-58 to +63 degrees). The defocus difference along the nanotube in such experimental conditions can easily be quite larger than the depth of focus. If the nanotube examined in this work was not entirely free of bends and its axis was not almost perpendicular to the rotation axis, the defocus difference along the nanotube would be quite larger than the depth of focus (e.g. quite larger than 10 nm in some tilting angles). Such a large defocus difference across the image affects the accuracy of the atom position determination in the reconstructed ptychographic phase images. The authors dealt with this problem by divided the 4D datasets in few slices along the nanotube and

performed reconstruction for each slice separately as noted in the manuscript: “For the best reconstruction quality of projections where the nanotube is tilted along the beam direction, we reconstruct slices along the nanotube separately with different probes”. The authors should provide further information regarding this procedure to demonstrate that the defocus difference along the nanotube has been carefully considered in the phase reconstruction. For example what was the size of those slices and how the sudden change in the defocus between slices affects the accuracy of the reconstructed phase?

We agree that such additional details could be helpful to the reader. The change in defocus from chunk to chunk is minute, less than a quarter of the depth of field as we have discussed in the comment below. We have now added more information on the chunking mentioned in the methods. The text now reads:

For the best reconstruction quality of projections, we consecutively reconstruct chunks of the nanotube, and feed the probe from the last reconstructed chunk in the reconstruction of the next chunk. The chunking scheme is shown in Supplementary Fig. 11. Reconstruction without this scheme is also possible, see for example Supplementary Fig. 2 for a full field of view reconstruction.

In addition, the authors provided a single probe function for each dataset in Fig. 1 as well as Fig. S5 while they used multiple probes for some (or all) of their datasets. It is not clear that those probes are for the whole or a part of those datasets. Furthermore, could the authors explain the considerable change in the reconstructed probe function caused by tilting observed in Fig. S5? Does those changes related solely to the defocus variation? (which I don't think is the case) It seems that the other aberrations are changing to compensate the change in the defocus variation across the field of view. If that is the case, it can considerably affect the accuracy of the results.

The change in the real-space probe from tilt to tilt stems mostly from defocus. To visualize this better, we now provide the condenser-plane profiles in Fig. S12-15. From Fig. S12, it is clear that the higher-order aberrations mostly stay the same, while the defocus is varying from tilt to tilt. This is expected since we did not spend time or dose budget with precise focusing as required for atomic resolution ADF-STEM, but instead relied on the joint reconstruction of the probes.

The defocus variation is very small for this sample, around 2 nm. We have added a 3D visualization of the full volume to drive this point home. Supplementary Fig. 10 shows a 3D rendering of the full volume. It is clear that the z-position of the nanotube center changes only by around 2 nm.

b) Identification of atoms Although ptychography is a powerful technique to obtain information about the position of atoms in a structure, it is not a proper technique for identification of the type of atoms in a sample. Hence, the sample should be examined by some quantitative techniques (e.g. ADF) to support the atomic structure determined in this work. The authors provided a comparison of ADF vs iCOM vs ptychography (Fig. S2) at the electron dose which they used for their tomography experiment to demonstrate that ADF cannot be used to obtain useful information from the sample due to the low the signal-to-noise ratio in their experimental condition. However, the electron dose used in their experiment has been chosen for the sample to tolerate electron irradiation during several times of data collection from one area. I believe that it is possible to use much higher electron dose to collect ADF images from this sample which can be used to discriminate between Zn and Te atoms. Alternatively, they can obtain ADF images with a suitable signal-to-noise ratio by averaging over 30 ADF images from a single area using the same experimental conditions employed in this work.

Here we must respectfully disagree. Collection of a full, meaningful ADF-STEM tomogram of this structure would be extremely challenging, and quite likely unachievable using available methods. The structure is freestanding and focusing is only possible on the structure itself. Collecting a single ADF-STEM micrograph is of course possible, as shown in multiple publications cited in the article, but collecting a whole ADF-STEM tomogram tilt series would incur much additional dose before and during acquisition. Based on our experiments, we do not believe we could collect a sufficient number of damage-free projections at the dose required for ADF-STEM to reconstruct the 3D structure. Also, the bending of the CNT would affect ADF-STEM imaging quality somewhat more than ptychography. This is because the defocus of the ADF image cannot be changed post-acquisition, while the ptychographic reconstruction process fully deconvolves the probe aberrations including defocus. Reviewer 1 agrees with us. Furthermore, our structural determination is corroborated with our DFT calculations where full structural optimizations (that is allowed all forces to minimize to find a low-energy structure) predict a strikingly similar structure to that suggested in experiments.

Finally, it would be great if the authors could provide information on the best achieved resolution (2D and 3D) in the manuscript.

This is a good point. We have included an additional section in the manuscript for the resolution determination. (See below). We have also included Supplementary Figs. 16 and 17 showing mean experimental atom volumes and simulated and PSF-blurred atom volumes.

Resolution

We estimate the resolution similar to [1]. First, we extract the mean Zr and mean Te atomic volumes from the experimental reconstruction and upsample to 0.1 Å pixel size. Then we simulate Zr and Te atomic potentials with the abTEM [2] package by averaging 250 frozen phonons at 293 K, with standard deviation $\sigma = 0.085$ Å calculated from crystalline Zr [3] and convert them to the 3D transmission function. Subsequently, we solve an optimization problem that minimizes the difference between a Gaussian PSF-blurred transmission function and the mean experimental atom volume for Zr and Te atoms. The FWHM of the determined PSF follows approximately the Abbe resolution [1]. For the mean Zr atom, we determine a resolution of $d_{\perp} = 1.02$ Å perpendicular to the missing wedge and $d_{\parallel} = 2.26$ Å along the missing wedge direction, giving a mean 3D resolution of $d_{\text{Zr}} = 1.63$ Å. For the mean Te atom we determine a resolution of $d_{\perp} = 1.1$ Å perpendicular to the missing wedge and $d_{\parallel} = 2.26$ Å along the missing wedge direction, giving a mean 3D resolution of $d_{\text{Te}} = 1.68$ Å. The mean experimental, simulated, and optimized PSF-blurred volumes for Zr and Te are shown in Supplementary Figs. 16 and 17, respectively.

Furthermore, we have added a small section in the conclusion highlighting the potential of ptychography to reach sub-Angstrom 3D resolution in the near future.

The reconstructed volume achieves a resolution of 1.02 Å and 1.1 Å for the Zr and Te atoms, respectively, perpendicular to the missing wedge direction and 2.26 Å along the missing wedge direction. We expect that 3D phase-contrast reconstructions with sub-Å resolution will soon be possible with advancements in detectors, reconstruction algorithms, and experimental protocols.

Minor modifications

Minor modifications to the main text to reflect changes from Review Phase 1 on the main text: The number of atoms changed slightly from 575 to 580. The R-factor between experiment and simulation was improved to 16 percent due to frozen phonon modeling. The change is reflected in the main text now.

- [1] Z. Chen, Y. Jiang, Y.-T. Shao, M. E. Holtz, M. Odstrčil, M. Guizar-Sicairos, I. Hanke, S. Ganschow, D. G. Schlom, and D. A. Muller, Electron ptychography achieves atomic-resolution limits set by lattice vibrations, *Science* **372**, 826 (2021).
- [2] J. Madsen and T. Susi, The abtem code: transmission electron microscopy from first principles, *Open Research Europe* **1**, 24 (2021).
- [3] V. Sears and S. Shelley, Debye–waller factor for elemental crystals, *Acta Crystallographica Section A: Foundations of Crystallography* **47**, 441 (1991).

REVIEWER COMMENTS

Reviewer #4 (Remarks to the Author):

I am pleased with the thorough responses provided by the authors addressing the concerns raised in the previous review. However, there is one remaining aspect that requires attention. It would greatly enhance the clarity of the structural determination of the encapsulated phase if the authors add ADF images from their specimen. To provide further clarity regarding my previous comment about collecting ADF images, I would like to emphasize that my intention was not to suggest collecting a whole ADF tomogram. In fact, I suggested to collect ADF images from a single projection from the specimen. In such case, the authors can use much higher dose to obtain ADF images with a suitable signal-to-noise ratio. The aim of this approach is to ascertain whether the resulting data align with the suggested sample structure as outlined in the manuscript. For example, referring to the proposed structure for the encapsulated phase shown in Fig. 2(d) (and also in the attached .cif file), it is proposed that the outermost layer of the encapsulated phase exclusively comprises Te atoms. Therefore, ADF images can, at least, be used to validate this aspect (as the ADF signal can be used to discriminate between Zn and Te atoms). One of the ideal viewing direction for ADF imaging is along the "y" direction (y direction in the .cif file), which could prove the existence of the suggested Te-Te chains within the structure. I acknowledge that acquiring ADF data from specific viewing directions in this sample is not straight forward and I am not requesting the authors to provide an ADF image from a specific viewing direction. However, I am confident that if the authors collect ADF images from a few selected regions, they might be able to obtain images that can, at the very least, validate the composition of the outermost layer, confirming the presence of only Te atoms.

I am confident that the paper will be ready for publication without the need for additional rounds of revisions, if the authors choose to include at least an ADF image to show that the outermost layer of the encapsulated phase exclusively comprises Te atoms, or if they believe that the signal-to-noise ratio is not suitable for such analysis, they should demonstrate this by providing a simulated ADF image with an electron dose >30 times greater than the dose they employed to simulate Fig. S1(a).

Response to the Reviewers

(Dated: September 12, 2023)

We thank reviewer 4 for their detailed feedback. Below is our response to the remaining comment.

Thank you for your time!

Reviewer 4 (Remarks to the Author):

I am pleased with the thorough responses provided by the authors addressing the concerns raised in the previous review. However, there is one remaining aspect that requires attention. It would greatly enhance the clarity of the structural determination of the encapsulated phase if the authors add ADF images from their specimen. To provide further clarity regarding my previous comment about collecting ADF images, I would like to emphasize that my intention was not to suggest collecting a whole ADF tomogram. In fact, I suggested to collect ADF images from a single projection from the specimen. In such case, the authors can use much higher dose to obtain ADF images with a suitable signal-to-noise ratio. The aim of this approach is to ascertain whether the resulting data align with the suggested sample structure as outlined in the manuscript. For example, referring to the proposed structure for the encapsulated phase shown in Fig. 2(d) (and also in the attached .cif file), it is proposed that the outermost layer of the encapsulated phase exclusively comprises Te atoms. Therefore, ADF images can, at least, be used to validate this aspect (as the ADF signal can be used to discriminate between Zn and Te atoms). One of the ideal viewing direction for ADF imaging is along the "y" direction (y direction in the .cif file), which could prove the existence of the suggested Te-Te chains within the structure. I acknowledge that acquiring ADF data from specific viewing directions in this sample is not straight forward and I am not requesting the authors to provide an ADF image from a specific viewing direction. However, I am confident that if the authors collect ADF images from a few selected regions, they might be able to obtain images that can, at the very least, validate the composition of the outermost layer, confirming the presence of only Te atoms. I am confident that the paper will be ready for publication without the need for additional rounds of revisions, if the authors choose to include at least an ADF image to show that the outermost layer of the encapsulated phase exclusively comprises Te atoms, or if they believe that the signal-to-noise ratio is not suitable for such analysis, they should demonstrate this by providing a simulated ADF image with an electron dose ≥ 30 times greater than the dose they employed to simulate Fig. S1(a).

We have opted to simulate ADF-STEM images where the nanotube is tilted such that the Te-chain lies at the edge of the core structure and is shadowed by as few atoms as possible. We have added the following text and two figures to the Supplementary Information discussing the possibility of determining the Te chain chemistry from ADF-STEM images.

Using multi-slice image simulations, we consider the possibility of determining the chemistry of the Te chain by tilting the sample such that the Te chain is on the edge of the nanotube and shadowed by the smallest possible number of Zr or Te atoms. We consider the case where the 30-fold dose of a single projection as used in the tomographic experiment is now used to image the perfectly oriented sample without pre-exposure, with aberration-free imaging conditions and experimental parameters as in Table I. In the simulations, we respectively swap out the Te atoms in the Te chain with Zr atoms and perform two sets of simulations with 35 noise instantiations each. The mean of the 35 images is shown in Fig 1 b) and c), while in b) the Te atoms in the Te chain were swapped with Zr atoms. 1 a) shows the orientation of the model such that the Te chains are on the outer edge. We determine the mean and standard deviation of the pixels comprising the Te and Zr atoms. These are shown in Fig. 2. It can be seen that under these perfect experimental conditions, the Zr intensity error bar is 1.58σ from the Te mean, i.e. even under perfect conditions the decision between Te and Zr atom can be made only with less than 90 % confidence.

Real conditions in the microscope will decrease this number further. These conditions include: inability to align the nanotube to a perfect orientation without significant pre-exposure and therefore shadowing by other Zr and Te atoms, imperfect ADF-STEM detectors with Poisson-Gaussian noise, which was not modeled here, imperfect focusing, imperfect aberrations without significant pre-exposure.

We conclude that ADF-STEM experiments to determine the chemistry of the Te-chain are unlikely to yield further insights.

FIG. 1. ADF-STEM intensity of an experiment where the nanotube is rotated such that the Te chain is at the edge of the nanotube filling and shadowed by very few other Zr or Te atoms, with a fluence of $30 \cdot 1.72 \times 10^4 e/\text{\AA}^2$

FIG. 2. ADF-STEM intensity of an experiment where the nanotube is rotated such that the Te chain is at the edge of the nanotube filling and shadowed by very few other Zr or Te atoms, with a fluence of $30 \cdot 1.72 \times 10^4 e/\text{\AA}^2$

Given these simulation results and the expected low confidence of determining the Te chain chemistry from ADF-STEM observations, we do not think it is reasonable to try to perform these very time-intensive experiments.

REVIEWERS' COMMENTS

Reviewer #4 (Remarks to the Author):

I am pleased to acknowledge the responses provided by the authors. The paper is now in a state where it is suitable for publication. Congratulations to the authors on their dedication and efforts.